# The Aquaporin 3 Polymorphism (rs17553719) Is Associated with Sepsis Survival and Correlated with IL-33 Secretion

**DOI:** 10.3390/ijms25031400

**Published:** 2024-01-23

**Authors:** Dominik Ziehe, Britta Marko, Patrick Thon, Tim Rahmel, Lars Palmowski, Hartmuth Nowak, Alexander von Busch, Alexander Wolf, Andrea Witowski, Jolene Vonheder, Björn Ellger, Frank Wappler, Elke Schwier, Dietrich Henzler, Thomas Köhler, Alexander Zarbock, Stefan Felix Ehrentraut, Christian Putensen, Ulrich Hermann Frey, Moritz Anft, Nina Babel, Michael Adamzik, Björn Koos, Lars Bergmann, Matthias Unterberg, Katharina Rump

**Affiliations:** 1Klinik für Anästhesiologie, Intensivmedizin und Schmerztherapie, Universitätsklinikum Knappschaftskrankenhaus Bochum, 44892 Bochum, Germany; dominik.ziehe@rub.de (D.Z.); britta.marko@kk-bochum.de (B.M.); patrick.thon@rub.de (P.T.); tim.rahmel@rub.de (T.R.); lars.palmowski@kk-bochum.de (L.P.); hartmuth.nowak@kk-bochum.de (H.N.); alexander.von-busch@kk-bochum.de (A.v.B.); alexander.wolf@kk-bochum.de (A.W.); andrea.witowski@kk-bochum.de (A.W.); jolene.vonheder@ruhr-uni-bochum.de (J.V.); michael.adamzik@kk-bochum.de (M.A.); bjoern.koos@rub.de (B.K.); lars.bergmann@kk-bochum.de (L.B.); matthias.unterberg@kk-bochum.de (M.U.); 2Center for Artificial Intelligence, Medical Informatics and Data Science, University Hospital Knappschaftskrankenhaus Bochum, 44892 Bochum, Germany; 3Klinik für Anästhesiologie, Intensivmedizin und Schmerztherapie, Klinikum Westfalen, 44309 Dortmund, Germany; bjoern.ellger@klinikum-westfalen.de; 4Department of Anesthesiology and Operative Intensive Care Medicine, University of Witten/Herdecke, Cologne Merheim Medical School, 51109 Cologne, Germany; wapplerf@kliniken-koeln.de; 5Department of Anesthesiology, Surgical Intensive Care, Emergency and Pain Medicine, Ruhr-University Bochum, Klinikum Herford, 32049 Herford, Germany; elke.schwier@klinikum-herford.de (E.S.); dietrich.henzler@klinikum-herford.de (D.H.); thomas.koehler@ameos.de (T.K.); 6Klinik für Anästhesiologie, Operative Intensivmedizin und Schmerztherapie, Universitätsklinikum Münster, 48149 Münster, Germany; zarbock@uni-muenster.de; 7Klinik für Anästhesiologie und Operative Intensivmedizin, Universitätsklinikum Bonn, 53127 Bonn, Germany; stefan.ehrentraut@ukbonn.de (S.F.E.); christian.putensen@ukb.uni-bonn.de (C.P.); 8Marien Hospital Herne, Universitätsklinikum der Ruhr-Universität Bochum, 44625 Herne, Germany; ulrich.frey@elisabethgruppe.de; 9Center for Translational Medicine, Medical Clinic I, Marien Hospital Herne, University Hospital of the Ruhr-University Bochum, 44625 Herne, Germany; moritz.anft@elisabethgruppe.de (M.A.); nina.babel@elisabethgruppe.de (N.B.)

**Keywords:** sepsis, outcome, aquaporin-3, polymorphism, genotype, IL-33, rs17553719

## Abstract

Sepsis is a common life-threatening disease caused by dysregulated immune response and metabolic acidosis which lead to organ failure. An abnormal expression of aquaporins plays an important role in organ failure. Additionally, genetic variants in aquaporins impact on the outcome in sepsis. Thus, we investigated the polymorphism (rs17553719) and expression of aquaporin-3 (*AQP3*) and correlated these measurements with the survival of sepsis patients. Accordingly, we collected blood samples on several days (plus clinical data) from 265 sepsis patients who stayed in different ICUs in Germany. Serum plasma, DNA, and RNA were then separated to detect the promotor genotypes of *AQP3* mRNA expression of AQP3 and several cytokines. The results showed that the homozygote CC genotype exhibited a significant decrease in 30-day survival (38.9%) compared to the CT (66.15%) and TT genotypes (76.3%) (*p* = 0.003). Moreover, *AQP3* mRNA expression was significantly higher and nearly doubled in the CC compared to the CT (*p* = 0.0044) and TT genotypes (*p* = 0.018) on the day of study inclusion. This was accompanied by an increased IL-33 concentration in the CC genotype (day 0: *p* = 0.0026 and day 3: *p* = 0.008). In summary, the C allele of the *AQP3* polymorphism (rs17553719) shows an association with increased *AQP3* expression and IL-33 concentration accompanied by decreased survival in patients with sepsis.

## 1. Introduction

Sepsis is a common life-threatening disease caused by the host’s dysfunctional response to infection and is accompanied by organ dysfunction [1]. The 2016 sepsis definition includes the use of an acute change in the Sequential Organ Failure Assessment (SOFA) score of ≥2 points to identify organ dysfunction [2]. In the clinic, the terms sepsis and septic must be differentiated and not confused. A septic patient may present with the same symptoms as a patient with sepsis, but the bacterial diagnosis may not be clear and a number of other pathogens need to be considered much more fully so that appropriate, pathogen-specific therapy can be initiated [3]. Rapid diagnosis and urgent intensive care treatments have partly reduced the mortality rate of sepsis in the past few years, but effective treatments for sepsis are still lacking [4]. Furthermore, the high complexity of the disease and the long-term treatment lead to multiple undesirable consequences, such as mental, physical, and functional disorders [4,5]. Accordingly, the identification of novel key proteins for sepsis treatment and further elucidation of the underlying molecular mechanisms are urgently needed.

Aquaporins (AQPs) represent interesting target genes for sepsis diagnosis and therapy [6,7,8]. AQPs are membrane channel proteins which can transport water and other small molecules selectively and efficiently [9,10]. Several studies in recent years found that dysregulated AQP expression induces harm to several important organs, including the heart, brain, kidney, and lung, during sepsis. This indicates that AQPs may represent novel biomarkers of sepsis [6,11]. However, aquaporin-3 (AQP3) could be of special interest, as it was demonstrated that AQP3 participates in the regulation of pulmonary and intestinal function as well as immune cells in sepsis [6,8,12,13]. AQP3 is a 30 kDa hydrophobic protein and is part of the aquaporin family [12]. AQP3 expression can be found in various parts of the intestine, regulates the proliferation and migration of intestinal epithelial cells, and is responsible for glycerol and hydrogen peroxide transport [13]. It is expressed and also has a direct function in immune cells [11,14,15]. The role of AQP3 described in immune systems is predominantly its function in T-cells [16,17]. Here, AQP3 regulates T-cell migration by mediating the transport of hydrogen peroxide [18]. As we have also demonstrated in the past, a polymorphism in the *AQP3* gene is associated with T-cell migration [19], and altered T-cell migration could play a role in sepsis patients. Furthermore, AQP3 mediates both chemokine production and T cell trafficking [20].

Previous studies in murine models of sepsis showed that the AQP3 expression level is downregulated after septic intestinal injury. Treatment with glycerol, which can substitute AQP3 function, can ameliorate the intestinal injury and improve sepsis survival [21]. The results verified tentatively that intestinal AQP3 expression plays an important role in protecting the intestine during sepsis [21]. Another study demonstrated that AQP3 participates in the regulation of pulmonary vascular permeability after sepsis, and the antioxidant Ss-31 has a protective effect on pulmonary vascular permeability by downregulating the expression of AQP3 and inhibiting reactive oxygen species production [22]. It seems that AQP3 has different effects depending on the tissue in sepsis. On the one hand, AQP3 protects the intestine [8], and, on the other hand, it damages pulmonary vascular permeability during sepsis [6]. Moreover, AQP3 seemed to improve the outcome in a sepsis mouse model [8]. In addition, AQP3 seems to be involved in the formation of the inflammasome [23], which is important in sepsis pathophysiology [24,25,26]. However, it is unclear whether AQP3 has an impact on the outcome of sepsis patients. We have already described an aquaporin 5 (AQP5) promoter polymorphism with an impact on survival [27]. Here, we aim to investigate whether *AQP3* promoter polymorphisms play a similar role and correlate with sepsis survival.

The aim of this study is to investigate the polymorphism (rs17553719) and the expression of AQP3 in sepsis patients and correlate these measurements with the outcome of sepsis patients and the release of several cytokines.

## 2. Results

### 2.1. Characterization of Sepsis Patients

The baseline characteristics of sepsis patients ranked by their genotypes are shown in Table 1. A total of 265 sepsis patients were enrolled. The genotype distributions were 50.9% TT genotype, 42.3% TC genotype, and 6.7% CC genotype (Table 1), which was in Hardy–Weinberg equilibrium. The mean age was 65 years (±14), and 63% of the patients were male. At the time of study inclusion, 87% were receiving mechanical ventilation, and 28% were taking hydrocortisone. The patients did not differ in baseline characteristics, such as age, gender, Sequential Organ Failure Assessment (SOFA) score at study inclusion, or percentage of mechanical ventilation (*p* > 0.05; Table 1). In addition, neither the focus of infection, time in the ICU, nor the administration of hydrocortisone were different. However, the homozygote CC genotype showed significantly decreased 30-day survival (38.9%) compared to the CT (66.1%) and TT genotypes (76.3%) (*p* = 0.003; Table 1).

### 2.2. Survival Analysis of Sepsis Patients Ranked by AQP3 (rs17553719) Polymorphism

Thirty-day survival was significantly associated with *AQP3* (rs17553719) polymorphism genotypes (*p* = 0.003, Figure 1). Univariate Cox regression analyses revealed *AQP3* CC genotype status as the strongest prognostic factor for 30-day survival (Table 2).

In a univariate Cox regression model, several factors were identified to impact on 30-day survival (Table 2). Homozygous CC subjects had a high and approximately twofold greater risk of death (hazard ratio, 1.695; 95% CI, 1.233–2.330; *p* = 0.001) compared to TT/TC genotypes (Table 2). The impact on survival was even stronger (HR 1.73, *p* = 0.002; Table 2) when only the existence of any promotor polymorphism, irrespective of the specific type, was evaluated. Univariate analysis revealed other important factors for survival, which were age, interleukin-33 (IL-33), interleukin-6 (IL-6) concentration at study inclusion, and SOFA score at study inclusion (*p* < 0.05; Table 2). In a multivariate model, the impact of the *AQP3* genotype remained the strongest independent factor with a hazard ratio of 1.788, while the most significant impact could be seen for the SOFA score on study inclusion (*p* < 0.001; Table 2).

### 2.3. AQP3 Expression Analysis in Sepsis Patients

We analyzed the AQP3 expression in the patients’ blood samples to elucidate the biological background of the genotype-dependent effect on survival. *AQP3* mRNA expression was significantly higher and nearly doubled in CC compared to CT (*p* = 0.0044; Figure 2) and TT genotypes (*p* = 0.0180; Figure 2) on the day of study inclusion.

### 2.4. Cytokine Analysis in Sepsis Patients

We analyzed whether the polymorphism was correlated with the cytokine expression of IL1β, IFNα2, IFNγ, TNF-α, IL-33, IL12p70, MCP-1, IL-6, IL-10, IL-17a, IL-18, or IL-23 to further investigate the effect of the polymorphism in sepsis patients. A correlation between interleukin-33 (IL-33) (day 0: 0.284; *p* < 0.001; day 3: 0.196; *p* = 0.031), tumor necrosis factor α (TNF-α) (day 0: 0.244; *p* = 0.003; day 3: 0.222; *p* = 0.015), and interleukin-12 p70 (IL12p70) (day 0: 0.183; *p* = 0.028; day 3: 0.217; *p* = 0.017), as well as IL-23 (day 0: 0.169; *p* = 0.008) and the C allele carriers (CT and CC-genotypes), was detected. However, cytokine expression was increased in all C allele carriers, but a significant effect could only be detected regarding the IL-33, day 3 IL-12p70, and day 1 IL-23 cytokine concentrations (Figure 3). The IL-33 concentration in plasma samples was nearly doubled in C allele carriers (day 0: *p* = 0.0110 and day 3: *p* = 0.0161; Figure 3). IL12p70 was increased in C-allele carriers (day 3; *p* = 0.0328), and IL-23 was increased on day 1 (*p* = 0.0366; Figure 3).

## 3. Discussion

We demonstrate with this study that the C allele of the *AQP3* rs17553719 polymorphism is associated with decreased 30-day survival in sepsis. Furthermore, the *AQP3* polymorphism represented an independent—and the most important—prognostic factor for survival in our sepsis patients. The estimated hazard ratio of nearly 2 for the CC genotypes compared with the TT genotype suggests that the C allele of the *AQP3* polymorphism may have important effects on AQP3 expression in sepsis. Hence, AQP3 expression might have a potential relevance in sepsis. This assumption is supported by the fact that C allele carriers had nearly double the initial *AQP3* expression compared to T allele carriers.

In our analysis, the *AQP3* CC genotype was the strongest prognostic factor for sepsis lethality, with a hazard ratio of more than four in the multivariate model. Hence, the polymorphism could be used for the risk stratification of sepsis patients.

However, the underlying molecular and physiologic changes linking increased AQP3 expression to decreased 30-day survival cannot be clarified by our study. These alterations remain to be elucidated at a basic research level. However, we can make a few speculations. In a previous study, we described an AQP5 polymorphism which was associated with decreased AQP5 expression and increased 30-day survival [27]. Here, the C allele, which is associated with decreased AQP5 expression and increased survival in severe sepsis, is associated with a decreased neutrophil cell migration in vitro. Thus, the link between the AQP5-1364A/C polymorphism and sepsis survival could be due to AQP5 expression and its impact on neutrophil cell migration [28]. It was demonstrated mechanistically that the CpG methylation of the AQP5 promoter correlates strongly with the AQP5 A/C-1364 genotype, and, hence, this could represent the mechanistic explanation for the decreased AQP5 expression in C allele carriers [29].

AQP3 is involved in T-cell function, and a different AQP3 expression alters T-cell migration [19,30]. Hence, in our study, an altered T-cell function could impact sepsis survival. In addition, AQP3 seem to have effects on the activation of the NLR family pyrin domain containing 3 (NLRP3)-inflammasome [23]. Inflammasomes regulate proinflammatory cytokine production, which depends on cell regulatory volume mechanisms [23,26]. Here, AQP3 could be involved in the regulatory volume decreases in NLRP3-mediated inflammation [30,31], and it could affect nigericin-induced interleukin-1β (IL-1β) release by facilitating cellular K^+^ efflux [23]. In addition, as AQP3 also transports hydrogen peroxide, intracellular reactive oxygen species (ROS) rising with subsequent inflammasome activation could also be a possible mechanism [14,18].

Most of the measured cytokines, such as TNF-alpha and IL-6, were not different in the patient groups. It has to be kept in mind that some patients received immunomodulatory therapy, especially hydrocortisone, which might mask any effect of elevated levels of cytokines in some genotypes [32]. This might explain the missing differences. However, as hydrocortisone therapy was not different between the patient groups, we assume that this plays a minor role. Moreover, it is noteworthy that carriers of the CC genotype of AQP3 have a higher amount of IL-33 compared to the TT/TC genotype [33]. Previous studies described that IL-33 seemed to be protective in sepsis [34,35,36]. However, our univariate Cox regression analysis indicated that decreased AQP3-expression is associated with better survival. In addition, C allele carriers have decreased survival and increased AQP3 expression.

The limitations of this investigation should be mentioned. Unrecognized selection bias, inherent in many genetic association studies, cannot fully be excluded. Moreover, although all sepsis patients were treated with a rather standardized multimodal regimen, because of the multifactorial nature of this disorder, we cannot exclude the possibility that unknown and potentially confounding factors exist. Nevertheless, for the indication given, the study population was not small, and multivariate Cox regression analyses revealed *AQP3* polymorphism (rs17553719) as an important and strong independent prognostic factor for 30-day survival in severe sepsis. This underscores the potential relevance of AQP3 expression in severe sepsis, regardless of the mechanisms involved.

In conclusion, the C allele of the *AQP3* polymorphism (rs17553719) is associated with decreased survival in patients with severe sepsis. Future studies are required to replicate these observations and unravel the precise molecular mechanism by which the C allele of the *AQP3* polymorphism (rs17553719) influences survival in severe sepsis. It is our hope that this mechanism leads to the identification of novel treatment regimens that target AQPs or their expression.

## 4. Materials and Methods

### 4.1. Study Design and Cohort

The SepsisDataNet.NRW is a multicentric study, which is located in the German state of North-Rhine-Westphalia (German Clinical Trial Registry No. DRKS00018871; http://www.sepsisdatanet.nrw accessed on 21 January 2024). In this study, patients were prospectively included when they fulfilled the SEPSIS-3 criteria. The approval of the Ethics Committee of the Medical Faculty of the Ruhr-University Bochum (registration no. 5047–14) or the responsible ethics committees of each study center was obtained. This study was conducted in accordance with the revised Declaration of Helsinki, good clinical practice guidelines, and local regulatory requirements. Patients were recruited, after written informed consent was acquired, over a period from 1 March 2018 to 31 December 2022, at seven different ICUs of tertiary-care and university hospitals. Adult patients with a sepsis diagnosis within the previous 36 h according to the current SEPSIS-3 definition (suspected/proven infection and an increase in their SOFA score by two points or more) were eligible for inclusion.

The exclusion criteria were as follows:-Age below 18 years at the time of ICU admission;-Withdrawal or withholding of consent;-Withdrawal of treatment.

### 4.2. Collection of Blood Samples

Whole blood was drawn at enrolment and at days 3, 8, and 28 and transferred directly to the laboratory for further processing. Serum was collected after centrifugation of whole blood samples collected in S-Monovetten (Sarstedt, Nümbrecht, Germany). Upon centrifugation (4000× *g*) for 2 min, the serum fraction was removed and stored at −80 °C for cytokine quantification. DNA was extracted from whole blood samples collected in DNA exact tubes using the my-Budget Blood DNA Midi Kit (Bio-Budget Technologies GmbH, Krefeld, Germany). RNA was collected into Tempus Blood RNA tubes (Applied biosystems, Waltham, MA, USA) and isolated by a Tempus™ Spin RNA Isolation Kit (Applied biosystems).

### 4.3. DNA Genotyping

DNA samples were utilized for DNA genotyping after isolation. For genotyping of the *AQP3* rs17553719 polymorphism, the TaqMan SNP genotyping assay (catalog no. 4351379 C__26504424_20; Thermo Fisher Scientific, Wilmington, NC, USA) was utilized. Genotyping was performed using the Thermo Fisher Scientific TaqMan^®^ SNP Genotyping Master Mix (Thermo Fisher Scientific) and Bio-Rad CFX Connect Cycler Systems (Bio-Rad Laboratories, Feldkirchen, Germany) following a protocol of 95 °C for 10 min, 40 cycles of 95 °C for 15 s and 60 °C for 60 s.

### 4.4. RNA Isolation, cDNA Synthesis, and qPCR

A quantity of 1 µg total RNA, which was quantified by spectrometry, was utilized for cDNA synthesis using a High-Capacity cDNA Reverse Transcription Kit (Thermo Fisher Scientific) according to the protocol. qPCR reaction was performed utilizing *AQP3* primer: *AQP3* _Se: 5′-GGAATAGTTTTTGGGCTGTA-3′; *AQP3*_As: 5′-GGCTGTGCCTATGAACTGGT-3′ and ACTB as a reference gene, as described earlier [37]. qPCR reaction was carried out utilizing GoTaq^®^ qPCR Master Mix (Promega, Madison, WI, USA) with the BioRad CFX connect qPCR device.

### 4.5. Measurement of Cytokines in Serum

Thirteen cytokines were quantified in serum samples, which were drawn at admission. The LEGENDPlex^TM^ Human Inflammation Panel 1 (BioLegend, San Diego, CA, USA) was utilized as described in the manufacturer’s protocol. In short, for antigen capture, 25 µL serum samples were incubated with LEGENDPlex^TM^ beads, then washed and incubated with detection antibodies. Afterwards, the fluorescence was quantified in a flow cytometer (Canto II, BD Biosciences, San Jose, CA, USA). When the recorded concentration of a cytokine was below the lower limit of detection (LOD), the value was set to 0 pg/mL; additionally, if a value was recorded as higher than the upper LOD, it was set to the upper LOD.

### 4.6. Statistical Analysis

The baseline characteristics of the patients are reported as percentages for categorical variables and as means with SDs or medians with interquartile ranges (25th and 75th percentiles), as appropriate. Normal distribution was tested by the Kolmogorov–Smirnov test. Categorical variables were compared using McNemar’s or Fisher’s exact tests. Continuous independent variables were compared using the Student’s *t*-test or the Mann–Whitney U test. Differences between genotypes in time series measurements were determined via a mixed-effects analysis with a post hoc Kruskal–Wallis multiple comparisons test. Thirty-day survival was compared between the groups using a Kaplan–Meier analysis and the log-rank test. The independence of risk factors was evaluated using Cox regression analysis. A *p*-value of lower than 5% was considered significant. If not stated otherwise, the data are always presented as means ± standard deviations (SDs) or, in the case of non-normally distributed variables, as means ± interquartile ranges (IQRs). Correlation analysis was performed using Spearman or Pearson correlation analysis, as appropriate. All analyses were performed using SPSS (version 25, IBM, Chicago, IL, USA). GraphPad Prism 9 (Graph-Pad, San Diego, CA, USA) was used for the graphical presentations.

## 5. Conclusions

We report for the first time that 30-day survival in sepsis patients was significantly associated with *AQP3* (rs17553719) polymorphism genotypes. The survival for the homozygous TT-genotype was 76.3%, for the heterozygous TC-genotype it was 66.1%, and for the homozygous CC-genotype, the survival rate was only 38.9%. Thus, the CC genotype of the *AQP3* polymorphism represents an independent prognostic factor for survival in sepsis patients. Furthermore, the CC genotype of *AQP3* is accompanied by a higher expression of AQP3 and IL-33 in sepsis. Thus, the polymorphism could be mechanistically linked to sepsis survival by the IL-33 concentration and could represent a novel biomarker for sepsis. Future studies are required to replicate these observations and to unravel the precise molecular mechanism by which the C-allele influences survival in sepsis.

## Figures and Tables

**Figure 1 ijms-25-01400-f001:**
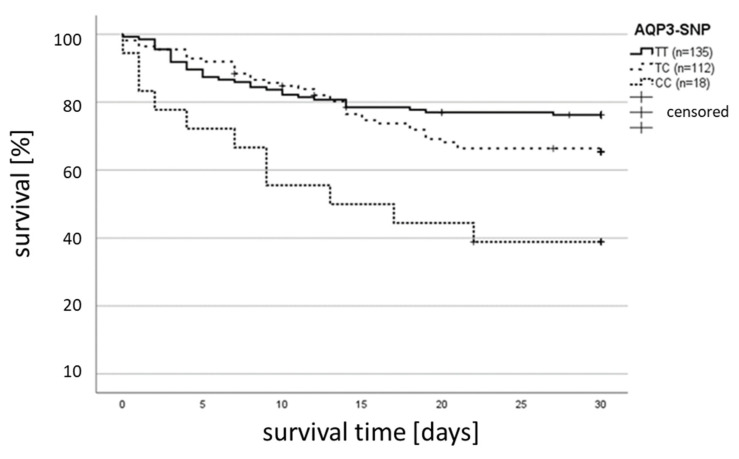
Thirty-day survival in patients with sepsis. Kaplan–Meier estimates were used to calculate probabilities of 30-day survival based on *AQP3* (rs17553719) polymorphism. A total of 265 patients were analyzed, and survival was compared by log-rank (Mantel–Cox) testing: 76.3% of TT-genotypes, 66.1% of TC-genotypes, and 38.9% of CC-genotypes survived; *p* = 0.003.

**Figure 2 ijms-25-01400-f002:**
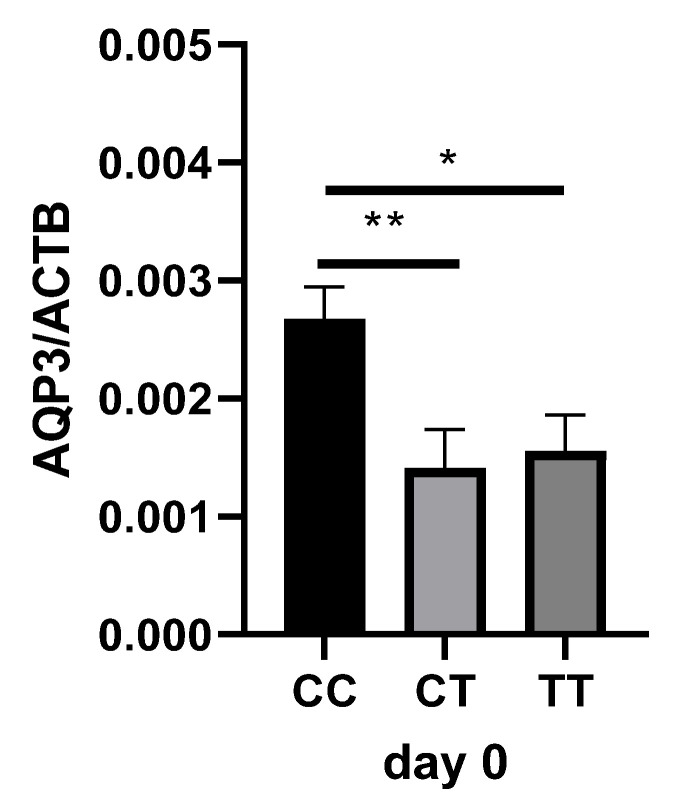
*AQP3* mRNA expression in relation to β-actin (ACTB) expression in whole blood (*n* = 84) of sepsis patients on the day of study inclusion in the ICU (day 0). The Kruskal–Wallis test was utilized for comparison of the groups with Dunn’s multiple comparisons test. * *p* < 0.05, ** *p* < 0.001; AQP3 = aquaporin-3; ACTB = β-actin.

**Figure 3 ijms-25-01400-f003:**
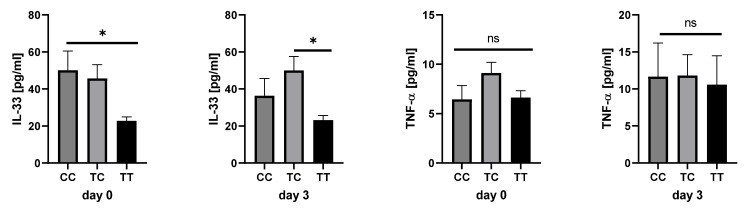
Plasma cytokine concentrations in sepsis patients (day 1, *n* = 122 (TT-genotype), *n* = 101 (TC-genotype), *n* = 15 (CC-genotype); day 3: *n* = 111 (TT-genotype), *n* = 87 (TC-genotype), *n* = 12 (CC-genotype)) ranked by the *AQP3* genotype. The cytokine concentration was determined by LEGENDPlex^TM^ ELISA. Kruskal–Wallis testing was performed with Dunn’s multiple comparisons test. * *p* < 0.05; *n* = number of patients with data. ns—not specified.

**Table 1 ijms-25-01400-t001:** Baseline characteristics of sepsis patients ranked by genotype.

	TT Genotype (*n* = 135)	TC Genotype (*n* = 112)	CC Genotype (*n* = 18)	*p*-Value
Gender male, n (%)	89 (65.9%)	66 (59.4%)	12 (68.8%)	0.542
Age in years, median [IQR]	65 [56–73]	64 [45–71]	67 [51–80]	0.726
SOFA score at day 1, median [IQR]	9 [5–12]	8 [5–10]	10.5 [5–14]	0.212
SAPS-II at ICU admission, median [IQR]	37 [28–41]	32 [22–35]	33 [26–41]	0.982
Mechanical ventilation, n (%)	88 (65)	75 (66.9)	10 (55.5)	0.781
Focus of infection, n (%)				0.458
Central nervous system	1 (0.7%)	2 (1.8%)	0 (0%)	
Lower respiratory tract	39 (28.9%)	41 (36.6%)	8 (44.4%)	
Skin and soft tissue	7 (5.2%)	5 (4.5%)	1 (5.6%)	
Genitourinary	5 (3.7%)	8 (7.1%)	0 (0%)	
Cardiovascular	6 (4.4%)	3 (2.7%)	0 (0%)	
Intra-abdominal	25 (18.5%)	11 (9.8%)	2 (11.1%)	
Musculosceletal	6 (4.4%)	2 (1.8%)	0 (0%)	
Unknown	46 (34.1%)	40 (35.7%)	7 (38.9%)	
Length of stay in ICU, median (days) [IQR]	8.9 [11.9]	6.9 [10.2]	8.9 [11.1]	0.733
Length of stay in hospital, median (days) [IQR]	10 [12]	7 [10]	9 [23]	0.428
30-day survival time, median (days) [IQR]	30 [3]	30 [18]]	11 [27]	0.003
30-day survival (yes), n (%)	103 (76.3%)	74 (66.1%)	7 (38.9%)	0.003
Hydrocortisone administration (yes), n = 181	25 (27.2%)	22 (28.6%)	5 (41.7%)	0.580

**Table 2 ijms-25-01400-t002:** Univariate and multivariate COX regression of 30-day survival of sepsis patients.

Co-Variable	Univariate			Multivariate		
	*p*-Value	HR	95% CI	*p*-Value	HR	95% CI
AQP3 genotype TT genotype	0.002			0.014		
TC genotype	0.119	1.454	0.908–2.327	0.023	2.231	1.116–4.459
CC genotype	<0.001	3.461	1.743–6.873	0.012	4.232	1.374–13.035
Age (per year)	<0.001	1.027	1.013–1.041	0.142	1.018	0.994–1.041
gender	0.124	1.320	0.927–1.880	0.540	1.239	0.623–2.464
IL-33 cutoff at study inclusion	0.021	1.665	1.080–2.567	0.406	0.704	0.307–1.613
TNF-α cutoff at study inclusion	0.004	1.842	1.209–2.807	0.179	1.601	1.936–7.699
IL-6 at cutoff study inclusion	<0.001	2.743	1.835–4.100	<0.001	3.861	1.936–7.699
SOFA score at study inclusion	<0.001	1.262	1.164–1.369	<0.001	1.280	1.153–1.420

## Data Availability

The data presented in this study are available on request from the corresponding author.

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
