# Peer review of "The Aquaporin 3 Polymorphism (rs17553719) Is Associated with Sepsis Survival and Correlated with IL-33 Secretion"

_ijms, 2024, doi:10.3390/ijms25031400_

Round 1

Reviewer 1 Report

Comments and Suggestions for Authors

This article is covering Aquaporin 3 polymorphism effects related to sepsis survival and correlated to IL 33 secretion. 

The team of 29 authors screened 265 sepsis patients and classified them into three genotypes TT, TC, and CC and ranked them with specific symptoms as listed in table 1.

The measurements of plasma cytokine concentrations for specific genotypes were determined by LEGEND Plex ELISA.

Additionally, article is concluded with only 32 literature references.  

This will constitute crucially important goals and novelty of this important paper. 

The following suggested changes and recommendations should be introduced before the publication of the manuscript.

It looks like that the authors are not clearly differentiate the term sepsis from septic.

Some literature data should be included into text in order to clarify the issue in order to correct the meaning and definition of sepsis.  Some example are below:

http://dx.doi.org/10.1016/j.ijid.2016.04.018 S

International Journal of Infectious Diseases 48 (2016) 118–119. 

See also:

https://www.thoracic.org/patients/patient-resources/managing-the-icu-experience/sepsis-severe-sepsis-and-septic-shock.php

Page 1, Line 20. Replace “ metabolic derangements”  with “metabolic acidosis”. which indeed has some effect on sepsis.Metabolic acidosis is frequently found in patients with severe sepsis. 

Page 1, Line 40. Replace “ reasonable” with  “urgent ”

Page 2. line 51. Replace “depict” with represents”. 

Page 2, Line 86. Replace “stratified” with “ranked” 

Page 3, Line 98. Replace “stratified” with “graded”.

Page 9, Line 283. Replace ““depict” with represents”. 

Page 9, Line 277. Conclusion, consist with only 6 lines and is short of specific justification of performed analysis and screening of 265 patients survival outcome. 

This section must be significantly expanded in order to illustrate the level of survival of sepsis patients 

The manuscript is of good quality and importance and is well written and edited in order to meet the standard for the articles published in International Journal of Molecular Sciences. 

Thus, I certainly recommend it for publication after the correction of these suggested minor changes. 

Author Response

Thank you very much for reviewing our manuscript. We hope that we have been able to answer everything to your complete satisfaction in the attached word document. 

Reviewer 2 Report

Comments and Suggestions for Authors

The study reports for the first time that 30-day survival in septic patients was significantly associated with AQP3 (rs17553719) polymorphism genotypes as mentioned the conclusion, however I have major concerns regarding the design and the final conclusion of the study

1- Sepsis can affect anyone, but people who are older, or have other health problems are at higher risk, how the authors cleaned the study from age factor or from other health problems that may interfere with the outcome

2- Treatment for sepsis requires medical care. Therapeutics like antimicrobials, anti inflammatory, intravenous fluids and careful monitoring, please indicate the therapeutic interventions in table 1, I am sure some of these therapeutics would prevent inflammatory mediators and keep low levels of cytokines which may mask any effect of elevated level of cytokines, the authors need to discuss

Comments on the Quality of English Language

Moderate editing of English language required

Author Response

(The authors gave the same response as above.)

Reviewer 3 Report

Comments and Suggestions for Authors

As an authors’ previous study suggested, organ failure in sepsis may be related to the aquaporin promoter polymorphism and the expression. The authors investigated the involvement of aquaporin 3 in septic patients. They found that the homozygote CC genotype of the AQP3 promoter was related to a decrease in 30-day survival, accompanied by increased AQP3 mRNA and IL-33 protein levels in blood. The results are interesting and could potentially lead to novel diagnosis methods and treatments for septic patients in the future.

Points to be addressed:

1. Line 26: It says ‘protein expression of AQP3’. However, only the mRNA expression data were presented. Please correct it.

 2. Line 129 (Figure 2 legend): What is ‘protein mRNA expression’? Please delete ‘and protein.’

 3. Figure 3: The CC, TC, and TT data should be presented separately in the same order as in Figure 2. Then, discuss the ‘C allele carriers.’ Please align the left and right graphs properly.

 4. It seems cytokines were measured to examine the ‘T-cell function.’ If so, please explain it in Introduction.

 5. Line 131: Only results of 3 cytokines were present. Please briefly mention the results of the other 10 cytokines measured.

 6. Line 248: Please specify the instrument with the company’s name used for qPCR.

Author Response

(The authors gave the same response as above.)

Round 2

Reviewer 2 Report

Comments and Suggestions for Authors

The manuscript can be accepted in its present form for publication

Comments on the Quality of English Language

minor editing of language is required